# Serum Advanced Glycation End Products and Their Soluble Receptor as New Biomarkers in Systemic Lupus Erythematosus

**DOI:** 10.3390/biomedicines12030610

**Published:** 2024-03-07

**Authors:** Irene Carrión-Barberà, Laura Triginer, Laura Tío, Carolina Pérez-García, Anna Ribes, Victoria Abad, Ana Pros, Jordi Monfort, Tarek Carlos Salman-Monte

**Affiliations:** 1Rheumatology Department, Hospital del Mar, 08003 Barcelona, Spain; ipirenaica@gmail.com (I.C.-B.);; 2Medicine Department, Medicine Faculty, Universitat Autònoma de Barcelona, 08193 Barcelona, Spain; 3Hospital del Mar Research Institute, 08003 Barcelona, Spain; 4Clinical Expertise Unit (UEC) in Systemic Autoimmune Diseases and Vasculitis, 08003 Barcelona, Spain

**Keywords:** advanced glycation end products, systemic lupus erythematosus, cardiovascular disease, activity index

## Abstract

It has been postulated that advanced glycation end products (AGEs) and their soluble receptor (sRAGE) may play a relevant role as inducers in the chronic inflammatory pathway in various conditions, among them, in immune-mediated diseases such as systemic lupus erythematosus (SLE). However, previous studies show conflicting results about their association with SLE characteristics and their usefulness as disease biomarkers. We aimed to study the association of specific serum AGEs (pentosidine, Nξ-(carboxymethyl)lysine (CML), Nξ-(carboxyethyl)lysine (CEL)), sRAGE levels and AGEs (specific serum AGEs and skin AGEs) to sRAGE ratios with various disease parameters, in order to clarify their potential as new biomarkers in SLE and to study their relationship with cardiovascular disease (CVD). To this aim, serum pentosidine, CML, CEL and sRAGE were measured via ELISA, and skin AGEs levels were measured by skin autofluorescence. Correlations of pentosidine levels with demographic and clinical data, indexes of activity, accrual damage and patient-reported outcomes were analyzed through multiple linear regression models, while correlations of the rest of the AGEs, sRAGE and AGE to sRAGE ratios (non-normal) were analyzed using both an OLS regression model and a GML. All of the analyses were adjusted for confounders. A total of 119 SLE patients were recruited. Serum AGEs and sRAGEs were significantly associated with SLE activity indexes and/or demographic or disease characteristics: pentosidine with pulmonary manifestations; CML with anti-dsDNA antibodies, IL-6, disease duration and non-Caucasian ethnicities; CEL with anti-dsDNA antibodies, IL-6 and accumulated number of manifestations; and sRAGE with male gender, photosensitivity and being on specific immunosuppressants. These results suggest that the AGE–sRAGE axis may serve as a novel biomarker for managing and prognosticating this disease. Its correlation with certain antibodies, demographics and disease presentations may indicate a distinct clinical phenotype associated with varying levels of AGEs and/or sRAGE. The significance of specific AGE/sRAGE ratios, introduced in this study for the first time, warrants additional investigation in forthcoming research. Our study did not confirm the link between serum AGEs and CVD, which merits further exploration through studies designed for this specific purpose.

## 1. Introduction

Systemic lupus erythematosus (SLE) stands as an autoimmune disorder marked by diverse clinical manifestations characterized by chronic inflammation and consequential organ damage. The intricate etiology of SLE involves a complex interplay of genetic, hormonal and environmental factors, along with the production of pathogenic antibodies and immune complex deposition [1]. 

Advanced glycation end products (AGEs) have been postulated to be pivotal participants in chronic inflammation [2]. AGEs, a diverse range of compounds, undergo intricate molecular processes resulting from the non-enzymatic interaction between reducing sugars, associated metabolites, peptides, proteins and amino acids. Under conditions such as aging, hyperglycemia and pro-oxidative states (e.g., diabetes mellitus, cardiovascular disease (CVD), chronic renal failure and neurological disorders), this interaction is enhanced, leading to the formation of protein adducts or cross-links [3], therefore increasing AGEs’ propensity to accumulate [4,5,6,7]. Systemic autoimmune diseases like SLE, which are characterized by inflammation as the hallmark of the disease, are among the factors that could potentially promote AGE formation. 

More than 20 AGEs have been identified in tissues, with Nξ-(carboxymethyl)lysine (CML) and pentosidine being the most studied due to their stability. Classical measurement methods involve chromatographic techniques and immunochemical methods such as enzyme-linked immunosorbent assay (ELISA) [8,9]. AGEs actively participate in promoting inflammation and reactive oxygen species generation through two primary mechanisms. First and foremost, glycation induces cross-links between modified proteins, leading to structural alterations and gradual dysfunction of cells and tissues. Additionally, AGEs interact with their receptors (RAGEs) expressed in various cells, including neutrophils, macrophages and T lymphocytes, triggering reactive oxygen species production and activating the NF-ϰB signaling pathway, ultimately contributing to inflammation [10,11,12,13]. 

RAGEs can be also found as a soluble form (sRAGE), acting as a decoy when binding to its ligands, which competitively bind to AGEs, inhibiting the proinflammatory processes mediated by the intracellular signal transduction of RAGEs [14,15]. While low sRAGE levels are commonly associated with inflammatory conditions, the paradoxical finding of elevated sRAGE levels in diseases like diabetes and chronic renal failure raises questions about their protective effect. Some authors propose exploring ratios between AGEs and sRAGE as potential universal risk markers for tissue damage, with better performance than AGE or sRAGE levels on their own [16,17,18,19]. 

Scarce previous research, with small sample sizes and simple statistics, have studied the relationship between AGEs and SLE, showing conflicting results (Appendix A). In terms of the association of AGEs with SLE characteristics, only three investigations have studied it: one finding no association with CML and Nξ-(carboxyethyl)lysine (CEL) [20]; the second one finding lower levels of pentosidine in patients with discoid lesions and photosensitivity, while positive direct Coombs test and malar rash were marginally associated (*p* = 0.09) with AGEs levels, inversely and directly, respectively [21]; and the third one reporting a direct association between the SLE disease activity index (SLEDAI) and AGE levels, but measured in plasma [22]. 

Serum sRAGE have been studied in deeper detail than serum AGEs, but also show discrepant results, as summarized in Appendix A. It is worth noting that the role of sRAGE in SLE is not clear, since although most studies have found lower sRAGE levels in SLE vs. healthy controls (HC) [23,24,25,26,27,28], as well as some inconsistent relationships with some SLE characteristics or indexes, three studies have described opposite results linking higher sRAGE levels with increased inflammation [20,29,30]. 

In SLE, the presence of accelerated atherosclerosis that cannot be fully explained by traditional risk factors for cardiovascular disease (CVD) is a well-recorded phenomenon [31]. Likewise, the role of AGE–RAGE has been suggested in atherosclerosis, with an increase in AGE production in the presence of several traditional cardiovascular risk factors (CVRFs) as hyperglycemia, aging and smoking, and some studies that additionally suggest that AGEs’ relation to CVD is independent from CVRFs [20,21]. Based on that, AGEs have been considered as a major CVRF, and have been proposed to be integrated into risk stratification of patients as well as in treatment decisions due to their pivotal role in the pathogenesis of cardiovascular arterial disease [22]. Some studies have suggested that increased levels of AGEs may contribute to the development of accelerated atherosclerosis in SLE, and therefore could be used as early markers for CVD in this pathology [32,33,34,35]. However, despite the association of AGEs or sRAGE with several CV factors in SLE, their role as early markers for CVD in this pathology is still unclear. 

This current study aims to address this research gap by investigating both serum AGE levels (CEL, CML and pentosidine), as well as sRAGE, in a multiethnic Spanish cohort of individuals with SLE. We try to answer some of the unmet needs through encompassing several specific goals; firstly, to explore correlations between these specific serum AGEs, sRAGE and the ratio between both serum and skin AGE to sRAGE concentrations and various demographic and SLE characteristics, including specific manifestations, activity or damage indexes, and patient-reported outcomes (PROs). Additionally, this research seeks to examine the association between AGEs and CVD, as well as CVRFs in the SLE population. The ultimate goal is to investigate the potential of AGEs as biomarkers for SLE in routine clinical practice. This includes their possible application for improving the monitoring and prognosis of SLE, as well as their potential as surrogate markers for assessing CVR in individuals with SLE. By addressing these objectives, the study aims to provide valuable insights into the role of the AGE–sRAGE axis in SLE and its potential clinical utility. 

## 2. Materials and Methods

### 2.1. Subjects

This cross-sectional study was conducted at the Hospital del Mar. Patients of all ages who visited the SLE outpatient clinic were randomly included. The selected patients met the 1997 American College of Rheumatology (ACR) [36] or the 2012 SLE International Collaborating Clinics (SLICC) classificatory criteria [37] for SLE, and accepted participation by signing informed consent. The exclusion criteria were pregnancy, diabetes mellitus (DM), treatment with glucocorticoids (GC) at a dose equivalent to prednisone > 20 mg/day, active malignancy and fibromyalgia.

It was estimated that a random sample of 97 individuals with SLE is sufficient to assess, with a 95% confidence interval (CI) and an accuracy of ±0.1 units, with the AGEs population mean values expected to have a standard deviation (SD) of about 0.5 units [35]. A diagram indicating the sample size used for each analysis is provided in Appendix A.

All of the patients signed the informed consent form to participate in the study. The protocol for our study was consistent with the provisions of the Declaration of Helsinki, and was approved by the ethics committee of the Hospital del Mar (CEIm-PSMAR 2018/7907/I).

### 2.2. Variables

A specific clinical visit was performed for this study. In this visit, demographic and clinical data were recorded, including indexes of activity, accrual damage and patient-reported outcomes (PROs) assessed through the recommended guidelines for each measurement. Furthermore, accumulated skin AGEs were also measured non-invasively in the skin with an autofluorescence reader (Age Reader Mu Connect^®^ DiagnOptics Technologies BV, Groningen, The Netherlands), as described previously in literature [38]. Briefly, the mean value was recorded from three consecutive AGE measurements taken from the ventral (anterior) surface of the forearm of each participant 10 cm below the elbow fold. The ratio between autofluorescence (measured between 420 to 600 nm) and the excitation light (emitted by a light source within the wavelength range of 320 to 400 nm) was recorded and expressed in arbitrary units (AU). Finally, a blood extraction was performed at the visit to determine the presence of autoantibodies, other biochemical compounds and specific serum AGEs and sRAGE. Antinuclear antibodies (ANAs) were determined by indirect immunofluorescence and considered positive if >1:80; anti-Ro60 and anti-Sm antibodies were determined by either multiplex immunoassay, being positive if titers > 1 antibody indexes or by blot, and considered positive for anti-double-stranded DNA (anti-dsDNA) antibodies by multiplex immunoassay with titers > 10 UI/mL. The measurement of serum AGEs is specified in Materials and Methods Section 2.3, while the other variables and their classifications are detailed in Appendix A. 

### 2.3. Assessment of Specific Serum AGEs

The ELISA method was used to evaluate the concentrations of three AGEs (pentosidine, CML and CEL) and sRAGE in the serum samples of each patient. During the study, the following ELISA kits were used according to the manufacturer’s instructions:-Human pentosidine sandwich ELISA kit (Cusabio Biotech Co., Ltd. Wuhan, China, CSB-E09415h); sensitivity 7.81 pmol/mL; precision measured as coefficient of variation < 8% (intra-assay), <10% (inter-assay).-Human CML sandwich ELISA kit (Cusabio Biotech Co., Ltd., Wuhan, China CSB-E12798h); sensitivity 15.6 pg/mL; precision measured as co-efficient of variation < 8% (intra-assay), <10% (inter-assay).-Human CEL sandwich ELISA kit (Cusabio Biotech Co., Ltd., Wuhan, China CSB-EQ027210HU); sensitivity 0.078 nmol/mL; precision measured as coefficient of variation < 8% (intra-assay), <10% (inter-assay).-Human receptor for AGEs, (RAGE/AGER) sandwich ELISA kit (Cusabio Biotech Co., Wuhan, China Ltd., CSB-E09354h); sensitivity 19.5 pg/mL; precision measured as coefficient of variation < 8% (intra-assay), <10% (inter-assay).

In the CEL assessment, some patients could not be included in the analysis due to the use of a different and not comparable ELISA kit, which has been discontinued.

### 2.4. Statistical Methods

The categorical data were described with absolute and relative frequencies, and continuous variables were displayed in terms of the mean (SD) or median (interquartile range) if non-normally distributed. Some continuous variables included in the final models were mean centered to facilitate interpretation. The assumptions of linearity, homoscedasticity and normality of the residuals were evaluated. If these premises were met, ANCOVA (analysis of covariance) multiple linear regression models were performed; however, when the assumptions could not be verified, mainly due to the right-skewed distribution of some variables, different multivariate regression models suitable for log-normal data were investigated with the aim of handling both heteroscedasticity and non-normality, and for estimating the absolute effect of each predictor. Finally, these multivariate analyses were performed using both ordinary least squares (OLS) regression models and generalized linear models (GLM) with gamma distribution and the identity link function. The assumptions of both models were evaluated assuming that the OLS model would be heteroskedastic in most of the analyses; therefore, the GLM model was used to verify and provide more evidence to the results obtained in the OLS model. The presence of influential points was also evaluated in each model through the Cook’s distance. All statistical analyses were carried out using R version 4.1.2.

In order to identify potentially confounding variables, in addition to a bibliographic review about previously reported factors related to AGEs, an exploratory analysis was performed using tertiles of the AGEs. This exploratory analysis was conducted using ANOVA tables, not only for the detection of confounding variables, but also to investigate associations between SLE patient characteristics and the level of each soluble AGE and sRAGE. For a better analysis, skewed variables of interest were categorized into tertiles or according to non-linear patterns and evaluated with general additive models. 

Associations with a *p*-value < 0.1 were considered significant and, if consistent, were examined individually. On the other hand, potentially confounding variables with statistically significant differences (*p* < 0.1), both between groups (characteristic yes/no) and AGE tertiles, were included in the final models to avoid spurious associations.

We also analyzed the associations with the ratios between specific serum AGEs or skin AGEs and sRAGE, as some authors determined that the ratios could be better biomarkers than AGE or RAGE levels on their own [18].

## 3. Results

The characteristics of the cohort of SLE patients are depicted in Table 1. Most of the patients were women (93.4%), mostly of Caucasian or Latin ethnicities, with low disease activity (65% in remission according to SLEDAI) and low damage (91% with SDI ≤ 2), and with a low number of CVRFs (61.5% with none) or CVEs (7.39% with ≥1 CVE).

### 3.1. Pentosidine

#### 3.1.1. Characteristics of SLE Patients According to Pentosidine Levels: Exploratory Analysis

A total of 117 SLE patients were included. Pentosidine met the normality and homoscedasticity premises (Appendix A), so the parametric statistical tests defined previously were performed. All of the variables that showed statistically significant differences according to pentosidine tertiles in the exploratory analysis are depicted in Table 2. Pentosidine was not found to be influenced by age or smoker status, so the analyses were not adjusted by any variable. The demographic characteristics and other SLE variables of interest are detailed in Appendix A.

#### 3.1.2. Correlations between Pentosidine and SLE Characteristics: Multivariate Analysis

SLE characteristics that were significant in the exploratory analysis and possibly related to pentosidine levels were tested in a model adjusted for previously selected confounding variables (see Materials and Methods). After adjustment, only the presence of pulmonary manifestations (lupus pneumonitis and shrinking lung syndrome) was strongly associated (Figure 1). Specifically, patients with lung involvement had pentosidine levels that were 1181.8786 (95% CI [507.4192; 1856.3379], *p* < 0.001) units higher than those without lung involvement. The model, which does not have any confounding factors, is provided in Appendix A.

### 3.2. CML

#### 3.2.1. Characteristics of SLE Patients According to CML Levels: Exploratory Analysis

A total of 117 SLE patients were included. CML presented a right-skewed distribution (Appendix A), so regression models suitable for log-normal data were performed, as defined in Materials and Methods. All of the variables that showed statistically significant differences according to CML tertiles in the exploratory analysis are depicted in Table 3. The demographic characteristics and other SLE variables of interest are detailed in Appendix A.

#### 3.2.2. Correlations between CML and SLE Characteristics: Multivariate Analysis

SLE characteristics that were significant in the exploratory analysis and possibly related to CML levels were tested in two models adjusted for the previously selected confounding variables (see Section 2). After adjustment, we found that non-Caucasian patients present higher values than Caucasian ones, and those positive for anti-dsDNA antibodies (≥11 IU/mL) also have increased CML levels. Finally, they also correlated with longer disease duration. These positive associations were significant in both OLS and GLM models. In addition, we also found that the 2nd tertile of anti-dsDNA antibodies (≥2 IU/mL) and 3rd tertile of IL-6 values [>3.24 pg/mL) presented higher CML levels than the 1st tertile, but both exclusively in the OLS model (Figure 2). The detailed models and their adjustments by confounding variables are provided in Appendix A.

### 3.3. CEL

#### 3.3.1. Characteristics of SLE Patients According to CEL Levels: Exploratory Analysis

A total of 91 SLE patients were included. The distribution of CML exhibited a right-skewed pattern (Appendix A), prompting the utilization of regression models tailored for log-normal data, as outlined in the Section 2. All of the variables that showed statistically significant differences according to CEL tertiles in the exploratory analysis are depicted in Table 4, not adjusted by any variable (*p*-value). The demographic characteristics and other SLE variables of interest are detailed in Appendix A.

#### 3.3.2. Correlations between CEL and SLE Characteristics: Multivariate Analysis

SLE characteristics that were significant in the exploratory analysis and possibly associated to CEL levels were tested in two models adjusted for the previously selected confounding variables (see Section 2). After adjustment, we found that the CEL levels correlated with anti-dsDNA antibodies, IL-6 levels and the number of accumulated manifestations throughout the disease (Figure 3a, Figure 3b, and Figure 3d, respectively). Furthermore, patients having ever had positive anti-dsDNA antibodies had significantly higher CEL levels (Figure 3c). These associations were found in both models except for one with the anti-dsDNA titers, which was only observed in the OLS linear regression model. Moreover, we found a correlation between CEL and CML levels (Appendix A). The detailed models and their adjustments by confounding variables are provided in Appendix A.

### 3.4. Serum Receptor for Advanced Glycation End Products (sRAGE)

#### 3.4.1. Characteristics of SLE Patients According to sRAGE Levels: Exploratory Analysis

A total of 119 SLE patients were included. The sRAGE distribution displayed a right-skewed pattern, as illustrated in Appendix A. Consequently, regression models designed for log-normal data, as described in the Section 2, were employed to analyze the dataset. All of the variables that showed statistically significant differences according to sRAGE tertiles in the exploratory analysis are depicted in Table 5, not adjusted by any variable. The demographic characteristics and other SLE variables of interest are detailed in Appendix A.

#### 3.4.2. Correlations between sRAGE Levels and SLE Characteristics: Multivariate Analysis

SLE characteristics that were statistically significant in the exploratory analysis and possibly associated with sRAGE levels were tested in the two models adjusted for the previously selected confounding variables (see Materials and Methods). After adjustment, we found that sRAGE levels were higher in women and in patients having ever had photosensitivity as an SLE symptom, as well as in those on biological disease-modifying antirheumatic drugs (bDMARD), which in our cohort included rituximab or belimumab, or mycophenolic acid (Figure 4). All of the associations were found in both models except for male gender, which was only found in the OLS linear regression model. The detailed models and their adjustments by confounding variables are provided in Appendix A. 

### 3.5. Ratios of Advanced Glycation End Products/Serum Soluble Receptor for Advanced Glycation End Products (AGEs/sRAGE)

#### 3.5.1. Characteristics of SLE Patients According to Skin AGEs/sRAGE or Specific Serum AGEs/sRAGE

All of the statistically significant associations in the univariate analysis are depicted in the Appendix A: Pentosidine/sRAGE in Appendix A, CML/sRAGE in Appendix A, CEL/sRAGE in Appendix A and skin AGEs/sRAGE in Appendix A.

#### 3.5.2. Correlations between Skin AGEs/sRAGE or Specific Serum AGEs/sRAGE and SLE Characteristics: Multivariate Analysis

After adjustment for confounding factors, we found several SLE characteristics that were associated with different serum AGE to sRAGE ratios, in one or both of the models. The pentosidine/sRAGE ratio was higher in those patients not following bDMARD treatment or having ever had anti-Ro52 antibodies (Figure 5). Regarding CML/sRAGE, non-Caucasian patients as well as patients showing SLICC/ACR damage index (SDI) ≥ 2 densitometric osteoporosis, or those on dyslipidemia drugs, presented higher ratios (Figure 6). CRP and IL-6 levels had a positive correlation with the CEL/sRAGE ratio, with those showing pathological IL-6 values displaying significantly higher ratios (Figure 7). Finally, the skin AGEs/RAGE ratio was lower in women and in those patients with disease duration > 16 years (3rd tertile) compared to those with disease duration < 5 years (1st tertile) (Figure 8). The detailed models and their adjustments by confounding variables are provided in Appendix A (pentosidine/sRAGE), Appendix A (CML/sRAGE), Appendix A CEL/sRAGE and Appendix A (skin AGEs/sRAGE).

## 4. Discussion

In this study, we observed that both the studied serum AGEs, including pentosidine, CEL and CML, along with their receptor, sRAGE, and the calculated ratios involving the latter, present associations with relevant clinical characteristics and indexes in SLE.

Concerning pentosidine, we only found one significant association; a nearly 80% increase in pentosidine levels were observed in patients with SLE pulmonary manifestations, which in our cohort only comprised shrinking lung syndrome and lupus pneumonitis, while pleuritis was considered inside the serositis term. Only one previous study analyzed the relationship between pentosidine and SLE characteristics, but they did not assess pulmonary manifestations because they only collected the ones included in the SLE classificatory criteria [21]. They found, however, lower levels of pentosidine in patients with discoid lesions and photosensitivity that we could not confirm in our cohort. Nevertheless, several characteristics were very different: in their cohort compared with ours, 37% of their patients were from African descent, while in ours < 10% were from an ethnicity different from Caucasian or Hispanic, overlapping with other inflammatory conditions, which was an exclusion criterion in theirs. The mean disease duration of their cohort was 24 months, while ours had a remarkably longer disease duration (only 41% of patients had a disease duration under 5 years). RAGE has been described to be constitutively highly expressed in the lung [39,40], and importantly linked to lung inflammation in several lung diseases [41]. Specifically, pentosidine has been associated with the progression to metastases in lung cancer and with asthma (measured in sputum) [42], where its role as a biomarker of a reduced response to bronchodilator treatment has been proposed [43]. Based on that physiological link and on the statistically strong association with these specific pulmonary symptoms in our study, pentosidine may represent a strong predictor of these infrequent but serious manifestations, and a useful tool in their monitoring.

When analyzing CML and CEL, we found similar associations with different SLE serological characteristics. These common results make sense, since we found a positive correlation between their levels, as has been reported in a previous study performed in HC [44]. Levels of both showed a positive correlation with anti-dsDNA antibodies and IL-6 (evaluated as tertiles in the case of CML, or as continuous variables in the case of CEL). We consider that the results in relation with CEL levels are more consistent with the cut-off used in clinical practice, as the CEL increase depends directly on anti-dsDNA antibodies and IL-6. In the case of CML, the tertiles did not match values considered positive (for anti-dsDNA antibodies) or pathogenic (for IL-6). Having taken into account that normal IL-6 values are considered < 7 pg/mL, and that the 3rd tertile includes both normal and abnormal values, we reassessed the association, splitting the sample into those with high values of IL-6 (>7 pmg/L) vs. normal (<7 mg/dL) values; however, we did not find differences between the groups, which makes the association difficult to interpret. Something similar happens in the case of anti-dsDNA antibodies, where both pathological values (>10 IU/mL, 3rd tertile) and the highest values in the non-pathological range [2–10 IU/mL, 2nd tertile] were associated with increased CML levels—the latter only according to the OLS model. Even though CML values included in the 2nd tertile are not considered positive, they are closer to being pathological if we consider anti-dsDNA values as a continuum, perhaps initiating a rise in their levels, a fact that has been associated with an increased risk of flares [45]. In addition, other associations with CML were also found. For each year of disease duration, CML levels increased by 1.7%; non-Caucasian patients showed CML levels almost 50% higher than Caucasian patients, and patients suffering from densitometric osteoporosis not associated to GC’ intake also showed increased CML levels (34.2%). Regarding CEL, for each new manifestation that the patient presented throughout the course of the disease (evaluated according to the symptoms included in either the ACR or the SLICC SLE classificatory criteria) we found CEL increases of 8.2%, while patients that had ever presented positive anti-dsDNA also had higher CEL levels (23.8%). There is only one previous study in the literature that studied the relation between SLE characteristics and CML or CEL without finding any association with disease indexes or characteristics or the number of accumulated manifestations according to the 1990 ACR classificatory criteria [20]. Nevertheless, the study was conducted with a very small sample size (10 SLE patients and 10 HC), and both AGEs were determined through mass spectrometry and not ELISA, which makes it non-comparable to our research.

All of the above characteristics are known to be correlated with disease indexes. For example, anti-dsDNA antibodies [46] and IL-6 titers are correlated with disease activity, despite the current failure of IL-6 blockade therapies [47]. The number of manifestations is also correlated with activity and possibly with organ damage, while disease duration is associated with organ damage [48] and non-Caucasian ethnicities, particularly African American and Caribbean ethnicities, with both activity and organ damage [49]. The fact that CML and/or CEL correlate with all those indexes opens the door to their use as a new activity/damage/prognosis biomarker in SLE. 

With regards to sRAGE, we found a negative association with male gender, showing almost 40% lower sRAGE levels than females. On the other hand, patients having ever had photosensitivity or being on treatment with bDMARD or with mycophenolic acid at the time of the study presented higher sRAGE levels (corresponding to increases of 31.3%, 111.3% and 59.8%, respectively). As stated in the Section 1, there is still much to elucidate about which sRAGE levels (high or low) are associated with inflammation because there is evidence for both, making interpretation of the results conflicting. Assuming the mainstream theory that supports low sRAGE levels as being deleterious in SLE, the fact that we found lower levels in males is consequential, since males are known to have more severe extrarenal and renal diseases [50]. In the case of the positive association with photosensitivity, we have several hypotheses: Firstly, patients who are photosensitive tend to protect themselves more from ultraviolet radiation, a notorious trigger for both cutaneous and systemic flares in SLE [51]. Secondly, photosensitive patients are normally treated with drugs that are photoprotective like hydroxychloroquine, which is known to absorb ultraviolet light in the skin in a concentration-dependent manner; it has also been demonstrated to reduce mortality in SLE [52] by preventing flares and organ damage, and by having an effect in other comorbidities such as thrombosis or bone destruction [53].

Looking at previous evidence published on the topic, there is a lack of consistent results regarding sRAGE in SLE. Ene et al. did not study the association between sRAGE levels and disease characteristics, but when compared with HC, found that sRAGE decreased by 7.6% in a non-lupus nephritis (LN) group (*p* < 0.001), by 5.8% in an LN group (*p* < 0.001) and by 5.5% in a type IV LN (*p* < 0.001) group [24]. Lan et al. observed that sRAGE decreased in the proliferative types of LN (III and IV) and in patients with poor response to treatment (those who did not achieve partial or complete renal remission with cyclophosphamide and GC therapy) [54]. The authors reported that although the reason why lower AGEs levels are related to poor response to treatment is unknown, it has most likely to do with the NF-κB pathway, which is activated by AGEs and blocked by both GC [55] and cyclophosphamide [56]. However, they did not find an association between sRAGE and activity measured by SLEDAI (r = 0.12 (95% CI: −0.02454 to 0.2653, *p* = 0.11) or the activity or damage index in kidney biopsies. We did not specifically study associations with types of LN individually, the renal response to treatment or indexes in the renal biopsy, as we had a small sample size of patients with LN (8 patients), which probably prevented us from finding any associations with it. Other authors, like Bobek et al., found a correlation of sRAGE levels with C4 concentrations in 37 children with SLE, although not with other indirect parameters of activity like the erythrocyte sedimentation rate (ESR), C-reactive protein (CRP) or anti-dsDNA titers [25]. Nowak et al. did not find either an association between sRAGE and disease characteristics or SLEDAI-2K in 31 SLE vs. 26 HC cases [23]. Discrepancies between their cohorts and ours (for example, children vs. adults) and the small sample sizes in most of these previous studies could explain the differences in the associations found regarding our research. 

Concerning the association of sRAGE with taking bDMARD and/or mycophenolic acid, there is scarce literature about the effect of immunomodulatory/immunosuppressant drugs in sRAGE. There is one study that observed a decrease in serum levels of sRAGE and esRAGE (endogenous secretory receptor for AGEs, generated through alternative splicing of RAGE mRNA) by 32.4% (*p* = 0.004) when treating patients with multiple sclerosis with fingolimod for 12 months. They also observed a decrease in pentosidine serum levels by 41.3% (although not significant), together with a decrease (although not significant) in clinical relapses [57]. In another study performed by Gross et al. in renal transplant recipients, sRAGE levels were statistically significantly inversely associated in the multivariate linear regression analysis with treatment with mycophenolate mofetil (βst = −0.21, *p* < 0.001) [58]. Low sRAGE levels were also associated with a 2–3 times higher risk of mortality (*p* = 0.006). Azathioprine, on the other hand, was associated with higher levels of sRAGE (*p* = 0.02), despite azathioprine also being associated with taking mycophenolate mofetil (r = −0.58, *p* < 0.001). The authors concluded that the relationship between mycophenolate mofetil and sRAGE requires further investigation, an affirmation that we fully support. 

Despite these limited data showing an association between lower sRAGE and treatment with immunosuppressants, we found that patients being treated with specific immunosuppressants showed an increase in sRAGE. Our hypothesis for explaining these results is that patients treated with bDMARD or mycophenolate mofetil have less inflammation, as these treatments are more potent inhibitors of inflammatory pathways than other treatments used for less severe disease. This would be supported by the previously mentioned study that found an association between higher AGEs and azathioprine [58]. This association of sRAGE with certain immunosuppressants could have therapeutic implications, as those treatments could be used to modulate sRAGE levels as well as inflammation. However, in the current study, we could not assess in more depth the relationship between immunosuppressants and SLE or check our hypothesis, so future studies should be designed for this specific purpose. 

On a different note, this is the first study in the literature to analyze the ratios between specific serum or skin AGEs and sRAGE in SLE. We found a significant relationship between these ratios and several variables. A statistically significant positive association was observed with the presence of anti-Ro52 antibodies in the blood test results, in non-Caucasian ethnicities, in the SDI (only in the OLS model), as well as densitometric osteoporosis, taking dyslipidemia drugs, CRP and IL-6 values, IL-6 pathological values (>7 pg/mL), and male sex. A negative association was found with being on bDMARD treatment and disease duration. Due to the novelty of these analyses and, until further validation of these results, the purpose of this part of our study was solely exploratory, taking into account that some authors defend the highest suitability as biomarkers of the ratios over the molecules on their own [18]. 

Despite the known relationship between AGEs and atherosclerosis, we did not find any correlation between serum AGEs, sRAGE or the ratios and either CVRFs or cardiovascular events (CVEs). However, the *p*-values in some of the exploratory analyses were <0.1 and, considering that we have a small number of patients with CVE (N = 9), it is likely that our results are limited by a lack of statistical power, thus preventing us from drawing conclusions about the role of AGEs or sRAGE in CVR. Furthermore, we only assessed CVD through traditional CVRF or CVE, and did not perform additional tests like the intima–media thickness (IMT) of the common carotid artery measured by ultrasound [35], or the small artery elasticity measured via pulse-wave analysis using tonometric recordings of the radial artery [32]; both of these tests have been associated with skin AGEs levels in previous research. Nowak et al. [23] did not find either that serum CEL, CML or sRAGE levels influenced the presence of CVD in their analyses, but it is necessary to point out that 80.65% of SLE patients had CVD in their cohort and the sample size was small (n = 31), which could have influenced the ability to find differences between groups. No other research studied the association between serum AGEs or sRAGE with CVD; in some studies, it was even used as an exclusion criterion [24]. 

Our study presents several limitations. Firstly, due to the retrospective nature of the study, some data could not be retrieved, such as the cumulative dose of GC taken throughout the disease; thus, we were only able to assess the impact of GC through the dose taken at the time of the study. Likewise, the design makes it impossible to assess causality, which warrants future prospective studies. Secondly, and to clarify the effect of longstanding disease and therapy in AGEs levels, studies should be performed in newly diagnosed patients with short disease duration who are naïve to treatments. Other limitations are that we did not measure the total serum AGE levels but some specific AGEs on their own. The fact that some characteristics occurred at low frequencies may also have had an influence on the statistical power. 

Our research represents a pioneering study that analyzed, in a deep and methodical way, the AGEs–RAGE axis in SLE, associating it with a vast array of demographic and clinical characteristics. There is scarce literature on this area, and our research has several strengths like the large sample size compared to other previously published studies; multiple and detailed data retrieved; complex statistics; and a comprehensive analysis encompassing serum individual AGEs, sRAGE, as well as their ratios. To our knowledge, this is the first study to find an association between SLE activity parameters and some accrual damage indexes with CML, CEL and sRAGE. Also, it is the first to report the ratios of skin AGEs or serum AGEs to sRAGE in SLE. Furthermore, we described, for the first time, AGE (pentosidine, CML and CEL) and sRAGE associations with specific serological and clinical parameters that could define more precisely a specific phenotype of patients in whom these molecules may have a particularly meaningful contribution. Therefore, our results are innovative and indicative of the promising role of AGEs and sRAGE as low-invasive surrogate biomarkers of SLE disease activity, damage and specific manifestations.

The next steps in continuing to determine the role of AGEs or sRAGE as biomarkers of the disease would include validation of our findings in independent cohorts; determination of clinically useful cut-off values; assessment of the performance of the biomarkers; and, finally, validation in external cohorts. Prospective studies are necessary to be able to establish causality and temporality, reduce selection and recall biases, analyze changes in the biomarkers throughout the course of the disease, and to improve the identification and control of confounders. Studies should be designed for specific purposes, such as assessing AGEs’ role in cardiovascular disease, and be sufficiently powered to find statistically significant differences in each case. 

## 5. Conclusions

The correlation observed between some serum AGEs and sRAGE with SLE activity and/or damage markers suggests that the AGEs–sRAGE axis has a role as a new biomarker in this disease related to management and prognosis, which would have enormous implications in a field where knowledge of SLE is currently lacking. Furthermore, the association of AGEs or sRAGE with specific antibodies and disease manifestations may indicate a particular clinical phenotype related to specific higher/lower AGEs and/or sRAGE levels, unveiling another potential clinical use of these products. 

AGEs/RAGE ratios have been proposed by some authors as better universal markers than their individual components. In this research, we described, for the first time in SLE, skin AGEs and serum AGE to sRAGE ratios and their association with activity, damage and severity markers, antibodies, treatments and comorbidities. However, the role of these ratios in SLE requires further assessment in future studies.

Finally, we could not find an association between AGEs or sRAGE and CVD or CVRF, but our sample did present a low number of CVEs and specific tests for CV assessment, and detections of subclinical atherosclerosis or undiagnosed CVRF were not performed. Subsequent studies designed to focus on these aspects should be carried out to explore this relationship.

## Figures and Tables

**Figure 1 biomedicines-12-00610-f001:**
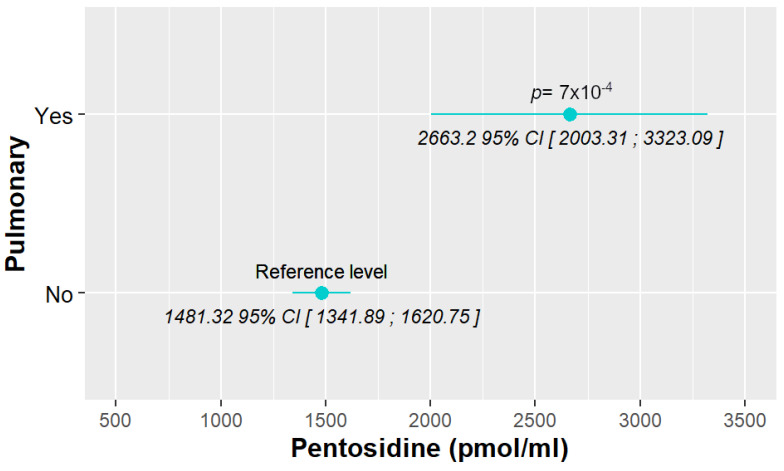
Associations between pentosidine levels and different SLE characteristics, with only pulmonary manifestations being significant.

**Figure 2 biomedicines-12-00610-f002:**
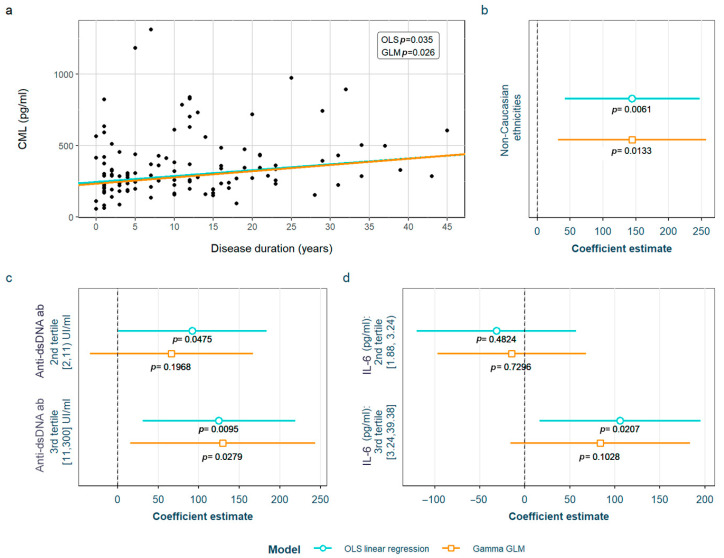
Statistically significant associations between CML and different systemic lupus erythematosus characteristics: (**a**) disease duration; (**b**) non-Caucasian ethnicities; (**c**) anti-dsDNA values; (**d**) IL-6 values. CML Nξ-(carboxymethyl)lysine; OLS: ordinary least squares; GLM: generalized linear model; IL-6: interleukin 6.

**Figure 3 biomedicines-12-00610-f003:**
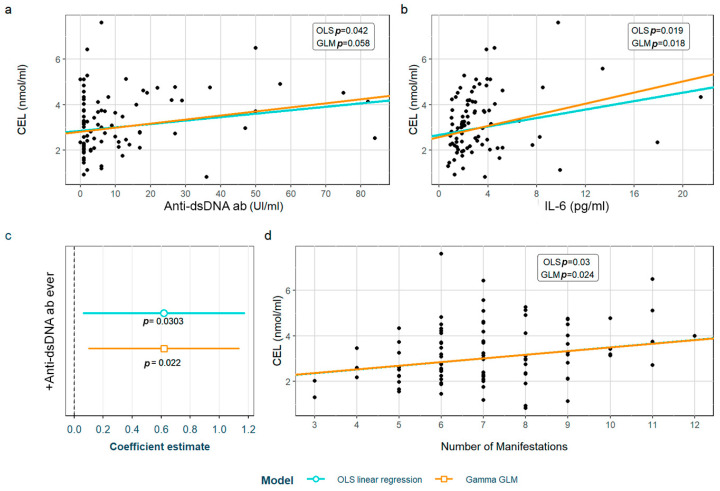
Statistically significant associations between CEL and different systemic lupus erythematosus characteristics: (**a**) anti-dsDNA values; (**b**) IL-6 values; (**c**) positivity of anti-dsDNA antiboides; (**d**) number of accumulated SLE manifestations throughout the disease. CEL: Nξ-(carboxyethyl)lysine; OLS: ordinary least squares; GLM: generalized linear model; IL-6: interleukin 6.

**Figure 4 biomedicines-12-00610-f004:**
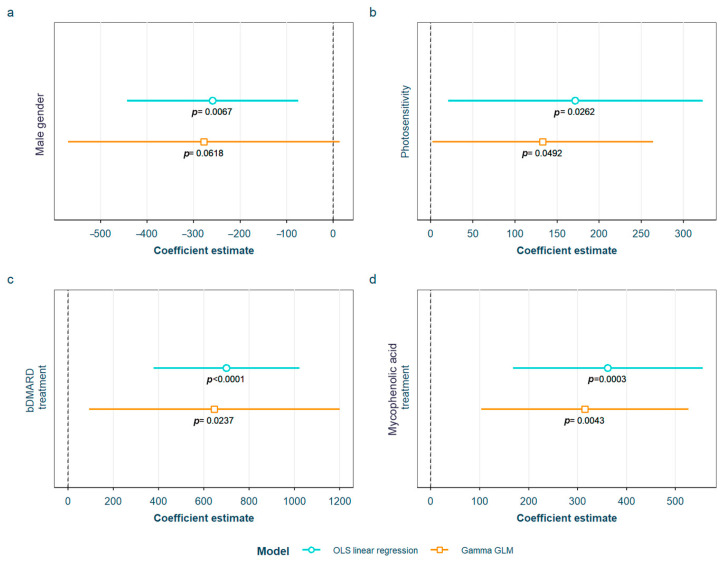
Statistically significant associations between the serum receptor for advanced glycation end products and different systemic lupus erythematosus characteristics: (**a**) male gender; (**b**) having ever had photosensitivity; (**c**) current treatment with bDMARD; (**d**) current treatment with mycophenolic acid OLS: ordinary least squares; GLM: generalized linear model; bDMARD: biological disease-modifying antirheumatic drugs.

**Figure 5 biomedicines-12-00610-f005:**
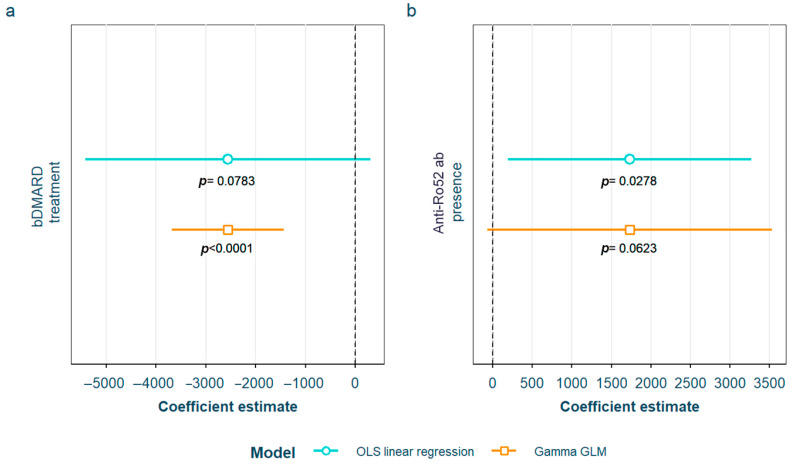
Statistically significant associations between pentosidine/sRAGE and different systemic lupus erythematosus characteristics: (**a**) current treatment with bDMARD; (**b**) positive anti-Ro52 antibodies sRAGE: soluble receptor for advanced glycation end products; OLS: ordinary least squares; GLM: generalized linear model. bDMARD: biological disease-modifying antirheumatic drugs.

**Figure 6 biomedicines-12-00610-f006:**
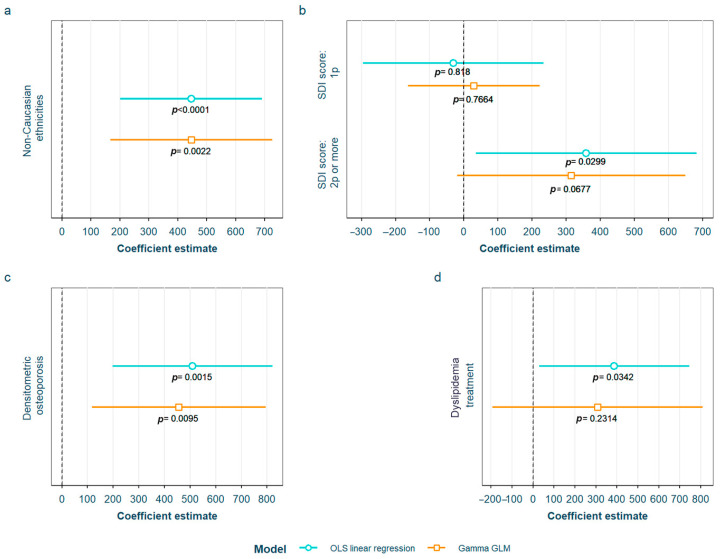
Statistically significant associations between CML/sRAGE and different systemic lupus erythematosus characteristics: (**a**) non-Caucasian ethnicities; (**b**) SDI score; (**c**) densitometric osteroporosis; (**d**) dyslipidemia drugs treatment CML: Nξ-(carboxymethyl)lysine; sRAGE: soluble receptor for advanced glycation end products; OLS: ordinary least squares; GLM: generalized linear model. SDI: systemic lupus erythematosus damage index.

**Figure 7 biomedicines-12-00610-f007:**
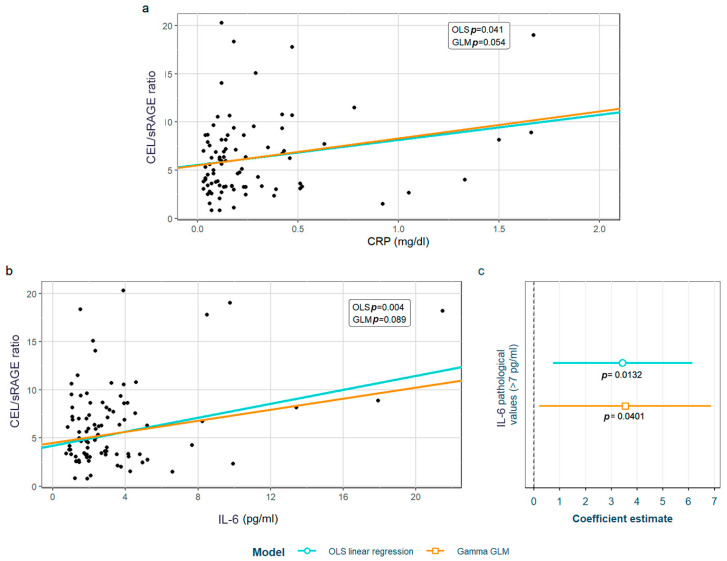
Statistically significant associations between CEL/sRAGE and different systemic lupus erythematosus characteristics: (**a**) CRP values, (**b**) IL-6 values; (**c**) pathological (>7 pg/mL) IL-6 values. CEL: Nξ-(carboxyethyl)lysine; sRAGE: soluble receptor for advanced glycation end products; OLS: ordinary least squares; GLM: generalized linear model. CRP: C-reactive protein; IL-6: interleukin 6.

**Figure 8 biomedicines-12-00610-f008:**
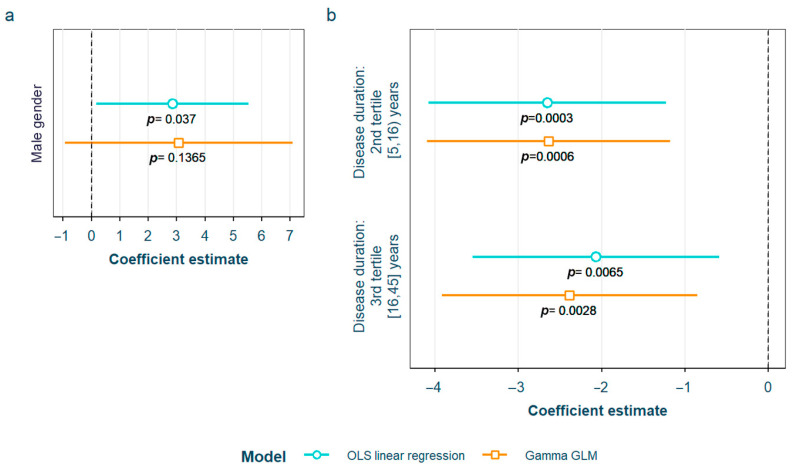
Statistically significant associations between skin AGEs/sRAGE and different systemic lupus erythematosus characteristics: (**a**) male gender; (**b**) disease duration in tertiles AGEs: advanced glycation end products; sRAGE: soluble receptor for advanced glycation end products OLS: ordinary least squares; GLM: generalized linear model.

**Table 1 biomedicines-12-00610-t001:** Demographic and disease characteristics of the SLE cohort. ESR: erythrocite sedimentation rate; CRP: C-reactive; protein; IL-6: antileukin 6; ANA: antinuclear antibodies; anti-dsDNA: anti-double-stranded; CH50, C3 and C4: complement CH50, C3 and C4; DAS28: disease activity score 28 joints; SLEDAI: SLE disease activity index; SDI: SLE damage index; PGA: physician global assessment; HAQ: health assessment questionnaire; VAS: visual analogue scale; FACIT: functional assessment of chronic illness therapy–fatigue scale; PtGA: patient global assessment; CVRF: cardiovascular risk factors (obesity = IMC > 30 kg/m^2^, arterial hypertension, dyslipidemia, chronic renal disease or hyperuricemia); CVE: cardiovascular events (angina, myocardial infarction, cerebrovascular accident, peripheral arterial disease, intestinal ischemia or ischemia of some other region); GC: glucocorticoids; cDMARD: disease-modifying antirheumatic drugs; bDMARD: biological DMARD. IS: immunosuppressants (includes treatment with methotrexate, leflunomide, tacrolimus, mycophenolic acid or mycophenolate mofetil acid, azathioprine, cyclophosphamide, cyclosporine, rituximab or belimumab).

Variables	All
	N = 122
Gender: Female	114 (93.4%)
Body mass index	25.4 (4.74)
Ethnicity	
Caucasian	81 (66.4%)
Latin	29 (23.8%)
Other	12 (9.84%)
Age	50.4 (14.9)
Smoker	32 (26.2%)
cDisease duration (years)	
0–5	50 (41.0%)
6–10	16 (13.1%)
11–20	33 (27.0%)
>20	23 (18.9%)
Serological Variables	
ESR *	11.0 [5.00; 20.0]
cCRP *	
[0.03, 0.12)	45 (37.2%)
[0.12, 0.28)	36 (29.8%)
[0.28, 3.92]	40 (33.1%)
cIL-6 *	
[0.63, 1.88)	36 (33.3%)
[1.88, 3.33)	36 (33.3%)
[3.33, 144.10]	36 (33.3%)
ANA+ *	112 (92.6%)
Anti-dsDNA+ *	4.00 [1.00; 13.0]
Anti-Ro52+ *	26 (21.8%)
Anti-Ro60+ *	45 (37.8%)
CH50 *	60.3 [51.8; 70.9]
C3 *	106 (22.3)
C4 *	19.8 (8.23)
SLE Activity Indexes	
cDAS28	
0—Remission	78 (65.0%)
1—Low activity	15 (12.5%)
2—Moderate activity	21 (17.5%)
3—High activity	6 (5.00%)
cSLEDAI	
Remission/Mild	71 (58.7%)
Moderate	39 (32.2%)
Severe	11 (9.09%)
SDI	0.00 [0.00; 1.00]
cSDI_3	
0–2	110 (90.9%)
3–4	8 (6.61%)
5–6	3 (2.48%)
PGA	2.00 [1.00; 3.00]
Patient-Reported Outcomes	
HAQ	0.38 [0.00; 0.88]
Patient pain VAS	2.00 [0.00; 6.00]
FACIT	17.5 [10.0; 27.0]
PtGA	2.75 [1.00; 5.00]
Comorbidities and Cardiovascular Disease	
Hypertension	26 (21.3%)
Dyslipidemia	12 (9.84%)
Cardiovascular disease	5 (4.10%)
Chronic renal disease	3 (2.46%)
Hyperuricemia	2 (1.64%)
Obesity	22 (18.0%)
CVRF > 0	47 (38.5%)
CVE	9 (7.38%)
CVRF and CVE > 0	48 (39.3%)
Treatments	
GC	30 (24.6%)
Current dose of GC	5.00 [2.50; 10.0]
Antimalarials	93 (76.2%)
cDMARD	19 (15.6%)
bDMARD	6 (4.92%)
Azathioprine	19 (15.6%)
Mycophenolic acid	20 (16.4%)
Tacrolimus	1 (0.82%)
cTreatment	
No IS	66 (54.1%)
IS	56 (45.9%)

* Indicates values according to the blood test performed in the study.

**Table 2 biomedicines-12-00610-t002:** Variables that showed statistically significant differences (*p*-value < 0.1) according to pentosidine tertiles in the exploratory analysis. * Indicates values according to the blood test performed in the study. SLE-DAS: SLE disease activity score; UPCR (mg/g): urine protein to creatinine ratio; OP: osteoporosis; CVE_SDI: cardiovascular events assessed in the SLE damage index (cerebral vascular accident, pulmonary infarction, angina or coronary bypass, myocardial infarction, venous thrombosis or infarction of the gastrointestinal tract); AGEs: advanced glycation end products; SD: standard deviation.

Variables	First Tertile [0, 1180)	Second Tertile [1180, 1594)	Third Tertile [1594, 4334]	*p*-Value
	N = 39	N = 39	N = 39	
Classificatory Criteria and Other Clinical and Serological Data				
Direct Coombs+ ever	4 (16.7%)	4 (21.1%)	1 (4.17%)	0.063
Pulmonary ever	0 (0.00%)	2 (5.13%)	3 (7.69%)	<0.001
Disease Activity Indexes				
SLE-DAS	4.18 [1.78; 7.28]	1.79 [1.20; 6.15]	2.53 [0.82; 4.86]	0.087
Serological Variables				
Total bilirubin *	0.32 [0.25; 0.48]	0.32 [0.26; 0.38]	0.35 [0.23; 0.41]	0.097
Hematuria *	0.00 [0.00; 0.00]	0.00 [0.00; 0.00]	0.00 [0.00; 0.00]	0.027
UPCR	84.6 [68.5; 133]	82.3 [63.5; 108]	74.7 [54.9; 90.7]	0.093
Comorbidities and Cardiovascular Disease				
Densitometric OP	4 (10.3%)	7 (17.9%)	7 (17.9%)	0.077
CVE_SDI				0.091
0	37 (94.9%)	34 (87.2%)	37 (94.9%)	
1	2 (5.13%)	4 (10.3%)	0 (0.00%)	
2	0 (0.00%)	1 (2.56%)	2 (5.13%)	
Treatments				
Tacrolimus	1 (2.56%)	0 (0.00%)	0 (0.00%)	0.093
Other AGEs				
Skin AGEs				0.065
<1SD	1 (2.56%)	1 (2.56%)	4 (10.3%)	
1SD-Means	4 (10.3%)	3 (7.69%)	6 (15.4%)	
Means	1 (2.56%)	2 (5.13%)	1 (2.56%)	
Means–>1SD	12 (30.8%)	10 (25.6%)	12 (30.8%)	
>1SD	21 (53.8%)	23 (59.0%)	16 (41.0%)	

**Table 3 biomedicines-12-00610-t003:** Variables that showed statistically significant differences (*p*-value < 0.1) according to CML tertiles in the exploratory analysis. “c” indicates variables that were categorized as previously stated in Section 2. * Indicates values according to the blood test performed in the study. PGA: physician global assessment; IL-6: interleukin-6; OP: osteoporosis; AGEs; advanced glycation end products; CEL: Nξ-(carboxyethyl)lysine.

Variables	First Tertile [57.6, 240)	Second Tertile [239.8, 383)	Third Tertile [382.9, 1555]	*p*-Value
	N = 39	N = 39	N = 39	
**Demographic variables**				
Ethnicity 3 categories				0.023
Caucasian	30 (76.9%)	29 (74.4%)	20 (51.3%)	
Latin	6 (15.4%)	7 (17.9%)	14 (35.9%)	
Others	3 (7.69%)	3 (7.69%)	5 (12.8%)	
Ethnicity 2 categories				0.006
Caucasian	30 (76.9%)	29 (74.4%)	20 (51.3%)	
Others	9 (23.1%)	10 (25.6%)	19 (48.7%)	
Disease-related variables				
Years of duration	4.00 [1.00; 14.5]	12.0 [4.00; 18.5]	12.0 [4.00; 21.0]	0.037
cYears of duration				0.088
0–5	22 (56.4%)	13 (33.3%)	12 (30.8%)	
6–10	5 (12.8%)	6 (15.4%)	5 (12.8%)	
11–20	9 (23.1%)	12 (30.8%)	11 (28.2%)	
>20	3 (7.69%)	8 (20.5%)	11 (28.2%)	
Tertiles years of duration				0.020
[0, 5)	21 (53.8%)	11 (28.2%)	10 (25.6%)	
[5, 16)	12 (30.8%)	14 (35.9%)	14 (35.9%)	
[16, 45]	6 (15.4%)	14 (35.9%)	15 (38.5%)	
Classificatory Criteria and Other Clinical and Serological Data				
Renal disease ever	0 (0.00%)	1 (2.56%)	7 (17.9%)	0.019
Disease Activity Indexes				
PGA	1.00 [1.00; 2.00]	2.00 [1.00; 3.00]	2.00 [1.00; 3.00]	0.094
Swollen joints	0.00 [0.00; 0.00]	0.00 [0.00; 0.00]	0.00 [0.00; 0.00]	0.093
Serological variables				
IL-6 tertiles *				0.050
[0.44, 1.88)	15 (40.5%)	12 (30.8%)	11 (28.9%)	
[1.88, 3.24)	13 (35.1%)	18 (46.2%)	7 (18.4%)	
[3.24, 39.38]	9 (24.3%)	9 (23.1%)	20 (52.6%)	
Comorbidities and Cardiovascular Disease				
Densitometric OP	5 (12.8%)	4 (10.3%)	9 (23.1%)	0.034
Treatments				
Dyslipidemia drugs	4 (10.3%)	1 (2.56%)	9 (23.1%)	0.004
Mycophenolic acid	2 (5.13%)	6 (15.4%)	12 (30.8%)	0.012
Glucocorticoids	8 (20.5%)	4 (10.3%)	18 (46.2%)	<0.001
Other AGEs				
CEL	2.45 [2.09; 3.71]	3.17 [2.47; 3.66]	3.99 [2.48; 4.68]	0.064

**Table 4 biomedicines-12-00610-t004:** Variables that showed statistically significant differences (*p*-value < 0.1) according to CEL tertiles in the exploratory analysis. “c” indicates variables that were categorized as previously stated in Section 2. * Indicates values according to the blood test performed in the study. “Treatment” divides patients into three groups according to the strongest immunosuppression they were taking at the moment of the study (only immunosuppressants, only antimalarials or neither (others)). “Treatment2” divides patients into two groups: taking or not taking immunosuppressants. CEL: Nξ-(carboxyethyl)lysine; SLE-DAS: SLE disease activity score; CRP: C-reactive protein; ESR: erythrocyte sedimentation rate; RV: reference value according to the laboratory; C3: complement C3; IL-6: interleukin-6; UPCR (mg/g): urine protein to creatinine ratio; IS: immunosuppressants (includes treatment with methotrexate, leflunomide, tacrolimus, mycophenolic acid or mycophenolate mofetil acid, azathioprine, cyclophosphamide, cyclosporine, rituximab or belimumab); “Treatment” divides treatment into three groups according to the strongest immunosuppression used (taking immunosuppressants, taking only antimalarials or not taking either (others)). AGEs: advanced glycation end products; CML: Nξ-(carboxymethyl)lysine.

Variables	First Tertile [0.823, 2.79)	Second Tertile [2.793, 4.56)	Third Tertile [4.564, 31.68]	*p*-Value
	N = 38	N = 37	N = 16	
Demographic variables				
Smoker	3 (7.89%)	8 (21.6%)	8 (50.0%)	0.087
Classificatory Criteria and Other Clinical Data				
Constitutional ever	3 (7.89%)	4 (10.8%)	1 (6.25%)	0.046
Photosensitivity ever	20 (52.6%)	27 (73.0%)	13 (81.2%)	0.089
Manifestations				0.006
3	2 (5.26%)	0 (0.00%)	0 (0.00%)	
4	2 (5.26%)	1 (2.70%)	0 (0.00%)	
5	7 (18.4%)	3 (8.11%)	0 (0.00%)	
6	9 (23.7%)	10 (27.0%)	2 (12.5%)	
7	8 (21.1%)	8 (21.6%)	5 (31.2%)	
8	6 (15.8%)	4 (10.8%)	3 (18.8%)	
9	3 (7.89%)	6 (16.2%)	2 (12.5%)	
10	0 (0.00%)	3 (8.11%)	1 (6.25%)	
11	1 (2.63%)	1 (2.70%)	3 (18.8%)	
12	0 (0.00%)	1 (2.70%)	0 (0.00%)	
Disease Activity Indexes				
cSLE-DAS				0.091
First tertile [0.82, 1.79)	19 (52.8%)	16 (47.1%)	2 (12.5%)	
Second tertile [1.79, 5.31)	6 (16.7%)	10 (29.4%)	5 (31.2%)	
Third tertile [5.31, 23.31]	11 (30.6%)	8 (23.5%)	9 (56.2%)	
Serological variables				
Glucose *	87.8 (12.4)	82.4 (8.96)	81.2 (7.69)	0.049
CRP *	0.12 [0.07; 0.28]	0.17 [0.11; 0.30]	0.16 [0.07; 0.54]	<0.001
ESR *	8.00 [4.25; 20.0]	10.5 [6.00; 15.0]	13.5 [7.00; 21.5]	0.054
Anti-dsDNA+ ever	23 (60.5%)	26 (70.3%)	14 (87.5%)	0.025
Anti-dsDNA+ *	2.50 [1.00;10.8]	5.00 [1.00; 13.0]	15.5 [1.75; 40.2]	<0.001
Anti-dsDNA > RV *	10 (26.3%)	10 (27.8%)	9 (56.2%)	0.018
Anti-dsDNA tertiles *				0.054
[0, 2)	16 (42.1%)	14 (38.9%)	4 (25.0%)	
[2, 11)	12 (31.6%)	12 (33.3%)	3 (18.8%)	
[11, 300]	10 (26.3%)	10 (27.8%)	9 (56.2%)	
Anti-dsDNA presence *	10 (26.3%)	10 (27.8%)	9 (56.2%)	0.018
Anti-Ro60+ ever	7 (18.4%)	19 (51.4%)	6 (37.5%)	0.097
Anti-Ro60 presence *	7 (18.9%)	17 (47.2%)	6 (37.5%)	0.086
Anti-Ro52+ ever	4 (10.5%)	12 (32.4%)	5 (31.2%)	0.060
C3 *	111 (24.3)	103 (19.2)	98.8 (20.5)	0.028
IL-6 *	1.98 [1.43; 3.77]	2.21 [1.81; 2.96]	3.92 [2.99; 6.03]	0.003
IL-6 > RV *	4 (10.8%)	2 (5.41%)	4 (25.0%)	0.002
IL-6 tertiles *				0.019
[0.44, 1.88)	16 (43.2%)	13 (35.1%)	1 (6.25%)	
[1.88, 3.24)	10 (27.0%)	15 (40.5%)	4 (25.0%)	
[3.24, 39.38]	11 (29.7%)	9 (24.3%)	11 (68.8%)	
UPCR *	82.2 [66.2; 119]	84.1 [63.0; 103]	71.3 [50.1; 121]	0.013
Treatments				
Mycophenolic acid	4 (10.5%)	8 (21.6%)	5 (31.2%)	0.007
NSAIDs	3 (7.89%)	4 (10.8%)	2 (12.5%)	0.038
Treatment				0.030
Others	6 (15.8%)	3 (8.11%)	0 (0.00%)	
Antimalarials	19 (50.0%)	13 (35.1%)	4 (25.0%)	
IS	13 (34.2%)	21 (56.8%)	12 (75.0%)	
Treatment2				0.009
Non-IS	25 (65.8%)	16 (43.2%)	4 (25.0%)	
IS	13 (34.2%)	21 (56.8%)	12 (75.0%)	
Other AGEs				
CML	281 [216; 374]	302 [248; 444]	464 [272; 711]	0.064

**Table 5 biomedicines-12-00610-t005:** Variables that showed statistically significant differences (*p*-value < 0.1) according to the serum receptor of advanced glycation end products tertiles in the exploratory analysis. “c” indicates variables that were categorized as previously stated in the Section 2. * Indicates values according to the blood test performed in the study. “Treatment” divides patients into three groups according to the strongest immunosuppression they were taking at the moment of the study (only immunosuppressants, only antimalarials or neither (others)). “Treatment2” divides patients into two groups: taking or not taking immunosuppressants. DAS28: disease activity score 28; ESR: erythrocyte sedimentation rate; VAS: visual analogic scale; bDMARDs: biologic disease-modifying antirheumatic drugs, IS: immunosuppressants (includes treatment with methotrexate, leflunomide, tacrolimus, mycophenolic acid, or mycophenolate mofetil acid, azathioprine, cyclophosphamide, cyclosporine, rituximab or belimumab).

Variables	First Tertile [122, 384)	Second Tertile [384, 671)	Third Tertile [671, 2797]	*p*-Value
	N = 40	N = 40	N = 39	
Demographic variables				
Gender: Female	35 (87.5%)	37 (92.5%)	39 (100%)	0.057
Classificatory Criteria and Other Clinical and Serological Data				
Photosensitivity ever	20 (50.0%)	29 (72.5%)	26 (66.7%)	0.022
Disease Activity Indexes				
DAS28	2.16 [1.49; 2.58]	2.10 [1.43; 3.24]	2.40 [1.57; 3.10]	0.050
cDAS28				0.008
0—Reference	31 (79.5%)	25 (62.5%)	21 (55.3%)	
1—Low Activity	2 (5.13%)	4 (10.0%)	8 (21.1%)	
2—Moderate Activity	4 (10.3%)	9 (22.5%)	7 (18.4%)	
3—High Activity	2 (5.13%)	2 (5.00%)	2 (5.26%)	
Serological variables				
ESR tertiles *				0.047
[2, 7)	13 (33.3%)	17 (42.5%)	12 (31.6%)	
[7, 17)	12 (30.8%)	10 (25.0%)	15 (39.5%)	
[17, 81]	14 (35.9%)	13 (32.5%)	11 (28.9%)	
Leukocyturia *	0.00 [0.00; 1.00]	0.00 [0.00; 1.00]	0.00 [0.00; 1.00]	0.022
Patient-Reported Outcomes				
Pain VAS	1.50 [0.00;5.00]	2.50 [0.00;6.12]	4.00 [0.00;6.00]	0.033
Comorbidities and Cardiovascular Disease				
APS	4 (10.0%)	1 (2.50%)	0 (0.00%)	0.097
Pain VAS	1.50 [0.00; 5.00]	2.50 [0.00; 6.12]	4.00 [0.00; 6.00]	0.033
Treatments				
bDMARDs	0 (0.00%)	2 (5.00%)	4 (10.3%)	0.002
Antimalarials	37 (92.5%)	27 (67.5%)	26 (66.7%)	0.009
Mycophenolic acid	7 (17.5%)	5 (12.5%)	8 (20.5%)	0.016
Azathioprine	2 (5.00%)	9 (22.5%)	7 (17.9%)	0.065
Glucocorticoids	13 (32.5%)	12 (30.0%)	5 (12.8%)	0.053
Treatment				0.016
Others	1 (2.50%)	6 (15.0%)	7 (17.9%)	
Antimalarials	24 (60.0%)	14 (35.0%)	13 (33.3%)	
IS	15 (37.5%)	20 (50.0%)	19 (48.7%)	
Treatment2				0.008
Non-IS	25 (62.5%)	20 (50.0%)	20 (51.3%)	
IS	15 (37.5%)	20 (50.0%)	19 (48.7%)	

## Data Availability

All data underlying the results are available as part of the article.

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
