# Peer review of "Serum Advanced Glycation End Products and Their Soluble Receptor as New Biomarkers in Systemic Lupus Erythematosus"

_biomedicines, 2024, doi:10.3390/biomedicines12030610_

Round 1
Reviewer 1 Report
Comments and Suggestions for Authors
An interesting cross-sectional Spanish study with the aim to analyze the potential interest of serum Advanced Glycation End-Products and their Soluble Receptors as potential additional biomarkers in SLE. Some comments to improve this interesting manuscript : while authors analyzed changes in tertile of values of AGEs and RAGE among clinical and serological characteristics of SLE patients and their correlations in multivariable analysis, simple correlations (with Pearson correlation index or non parametric Spearman correlation index ) between AGEs RAGE, AGEs/RAGE ratio and serological index of SLE activation (as anti-DNA double strand, C3 and C4 fractions ) and inflammatory bio-markers could be also of interest and increase the interest go AGE and RAGE for clinicians. Finally, in the discussion, authors should wisely beyond the pivotal findings of heir study as the ground of potential interest of these new biomarkers in SLE , describe their future steps for qualifying AGE and RAGE as additional biomarkers of interest in SLE especially for long term cardiovascular complications and they should insist on the need of longitudinal studies.
Comments on the Quality of English LanguageGlobally, good quality of the English
Reviewer 2 Report
Comments and Suggestions for Authors
The article presents the possible relationship between various biomarkers and Systemic Lupus Erythematosus, investigating novel markers that can be used in the prognostic and clinical follow-up of patients with SLE. It is well-written, with a systematic structure, and requires minor revisions:
- R48-49 rephrase
- R55-57 rephrase
- R78-79 rephrase
- Define “HC”, and “DM” when first using them in the text
- R207-210 - Delete the general information regarding the results section. Include general information from the present study (demographic information, stage of the disease, treatment, disease activity, etc.) Maybe include in the text also, a small table with general characteristics of the cohort, apart from the information that exists in the supplementary files.
- Please introduce the correct references for the sections you mentioned in the text (“Error! Reference source not found.”)
- R333 - provide the bDMARD used)
- R410-413 rephrase
- R452-453 rephrase
